# Immunocytometric Analysis of Oral Pemphigus vulgaris Patients after Treatment with Rituximab as Adjuvant

**DOI:** 10.3390/biom11111634

**Published:** 2021-11-04

**Authors:** Giulio Fortuna, Elena Calabria, Massimo Aria, Amerigo Giudice, Michele Davide Mignogna

**Affiliations:** 1Glasgow Dental School & Hospital, School of Medicine, Dentistry and Nursing, College of Medical, Veterinary and Life Sciences, University of Glasgow, Glasgow G2 3JZ, UK; giulio.fortuna@gmail.com; 2D.eb.RA. Mexico Foundation, Otomí #211, casi esq. P. Elías Calles Colonia Azteca, Guadalupe N.L., Monterrey 67150, Mexico; 3Federico Navarro Institute, School of Orgonomy “Piero Borrelli”, Corso Umberto I, 35, 80138 Naples, Italy; 4Department of Neurosciences, Reproductive Sciences and Dentistry, Federico II University of Naples, via Pansini 5, 80131 Naples, Italy; mignogna@unina.it; 5Department of Economics and Statistics, Federico II University of Naples, via Cinthia, Monte Sant’Angelo, 80126 Naples, Italy; aria@unina.it; 6School of Dentistry, Department of Health Sciences, Magna Graecia University of Catanzaro, 88100 Catanzaro, Italy; a.giudice@unicz.it

**Keywords:** rituximab, remission, oral pemphigus vulgaris, disease severity, OPV, desmoglein

## Abstract

**Background:** B-cell depletion therapy was demonstrated to be a valid and safe alternative as an adjuvant in oral-pharyngeal pemphigus vulgaris (OPV) patients. We aimed to assess its effects on anti-desmoglein (Dsg) 1 and 3 and leukocytes subsets profile in these patients’ population. **Methods and Materials:** We evaluated the immunologic profile of 10 OPV patients treated with RTX as adjuvant by using the ELISA testing for anti-Dsg-1 and -3 titers and the immunophenotyping for B and T-cell lymphocyte subpopulations and compared them with the PDAI score for clinical remission. **Results:** A significant difference in medians between baseline, end of RTX therapy, and 6 months after RTX therapy was observed in Dsg-3 titer (*p* < 0.001), in the CD8 (*p* = 0.009), and CD20 counts (*p* < 0.001). Multiple comparisons after Bonferroni adjustment confirmed such significant differences mainly between baseline and the end of RTX therapy and baseline and 6 months after RTX therapy. Only the anti-Dsg-3 titer at the end of RTX therapy demonstrated a slight positive correlation with the PDAI score at baseline (*p* = 0.046, *r =* 0.652). **Conclusions:** B-cell depletion adjuvant therapy in OPV patients demonstrated a significant impact on anti-Dsg-3 titer and B and T-cell lymphocyte subpopulations profile.

## 1. Introduction

Pemphigus Vulgaris (PV) is a potentially fatal autoimmune mucocutaneous blistering disease characterized by a suprabasal intraepithelial detachment [1]. This phenomenon, known as acantholysis, is caused by a humoral response against a broad spectrum of antigens responsible for cell-cell adhesions [1].

Although the first discovered family-antigen was a group of Ca^2+^-dependent molecules, desmogleins (Dsg) [2], over the past 20 years, more than 50 new non-desmoglein antigens have been identified as responsible for playing a role in PV pathogenesis, such as desmocollins 1 and 3, several muscarinic and nicotinic acetylcholine receptor subtypes, mitochondrial proteins, human leukocyte antigen molecules, or thyroid peroxidase [3].

First-line treatment of PV has always been based on conventional immunosuppressive therapy (CIST) with a high dose of corticosteroids (CS) and CS-sparing agents, such as azathioprine, cyclophosphamide, or mycophenolic acid, in order to reduce the total dose of CS and minimize mortality and morbidity [4,5]. However, prolonged use of these medications has been associated with significant unwanted side effects [5] and/or sometimes a failure in achieving a complete clinical and/or immunological remission [6].

The introduction of biologic agents, such as intravenous immunoglobulins G (IVIgG) and rituximab (RTX) used alone [7] or in combination [8], has dramatically changed the prognosis of PV patients, demonstrating a high safety and effective profile [9,10,11].

In pemphigus, RTX has been successfully used with both lymphoma protocol at a dose of 375 mg/m^2^ at weekly intervals for 4 weeks and rheumatoid arthritis protocol at a dose of 1000 mg twice at 2-week intervals [11,12]. A modified lymphoma protocol along with IVIgG [8] and a low dosage of RTX consisting of two doses of 500 mg each two weeks apart [13] have been reported.

Recently, it was demonstrated that RTX may represent a valid and safe alternative as adjuvant even in PV patients with oral-pharyngeal manifestations who were either non-responders or developed side effects to CIST, showing a low rate of relapses and side effects [14]. Unfortunately, the evolution of anti-Dsg-1 and -3 titers as well as the T and B-cell response after RTX therapy in OPV patients still remains unknown.

The primary endpoint was the evaluation of immunological variations in OPV patients after therapy with RTX, in terms of anti-Dsg1 and -3 and leukocytes subsets (CD4, CD8, CD20) variations, whereas the secondary endpoint was to determine any possible correlation between these variables and clinical remission.

## 2. Materials and Methods

### 2.1. Study Design and Patients

We performed a retrospective single-center study, analyzing data from OPV patients treated between 2013 and 2018, at the Complex Oral Medicine Unit of the Neurosciences, Reproductive and Odontostomatological Sciences, Federico II University of Naples. All patients provided their written informed consent for their therapeutic and personal data management before participating in the study. This study was approved by the local ethical committee of the University of Naples, Federico II (prot n. 69/19, 01/04/2019).

We included OPV patients aged 18 years or older if they met the following criteria: (1) clinical findings with active bullous and/or erosive lesions on the oral mucosa suggesting of PV; (2) histopathological (acantholysis) and immunological (direct immunofluorescence, and indirect immunofluorescence or ELISA test for antibodies (Ab) anti-Dsg 1 and 3) criteria for PV as previously described [4]; (3) classified as non-responders and/or developed side effects to CIST; (4) relapse during tapering CIST or after CIST, and thus were treated with rituximab as adjuvant therapy.

Conversely, the exclusion criteria encompass: (1) patients with mucocutaneous or cutaneous PV; (2) patients with concomitant severe systemic diseases, such as solid and/or non-solid neoplasm, myopathies, gastrointestinal, pulmonary, cardiac or renal diseases, or coagulation disorders; (3) patients with other concomitant autoimmune pathologies, such as systemic lupus erythematous, Sjögren syndrome, systemic sclerosis, which showed oral and/or cutaneous lesions and/or systemic manifestations, such as lichenoid lesions, granulomatous lesions, glomerulonephritis, endocarditis, pleuritis, myalgia, and abdominal complaints; (4) drug-addicted or alcoholic patients; (5) pregnant or breast-feeding women; (6) patients unable to provide their consent to the study.

Clinical data were collected as previously reported [14]; immunological data were collected as follows: anti-Dsg-1 titer, anti-Dsg-3 titer, CD4^+^, CD8^+^, and CD20^+^ percentage at baseline, at the end of RTX therapy, at complete clinical remission (CCR), at 6 months after the end of RTX therapy.

### 2.2. Anti-Dsg-1 and Dsg-3 IgG Enzyme-Linked Immunosorbent Assay (ELISA)

Circulating levels of anti-Dsg-1 and anti-Dsg-3 IgG antibodies were measured in serum samples by a commercially available enzyme-linked immunosorbent assay (ELISA) (MBL, Medical and Biological Laboratories, Nagoya, Japan) using a cut-off of 14 and 7 U/mL for both anti-Dsg 1 and anti-Dsg 3 antibodies, respectively.

The procedure was performed according to the manufacturer’s instructions. Microwell strips coated with recombinant purified Dsg-1 and Dsg-3 antigen were incubated with 100 μL of each diluted (1:101) patient’s serum, washed 4 times with PBS and Tween 20, then incubated with 100 μL of horseradish peroxidase-conjugated mouse monoclonal anti-human IgG at a 101× concentration, and washed again to remove all unbound antibodies. Afterward, 100 μL of substrate with 3,3′,5,5′-tetramethylbenzidine dihydrochloride/hydrogen peroxide (TMB/H_2_O_2_) was incubated in each well to produce a color change when reacted with the complexes previously formed. Finally, 100 μL of stop solution with 1.0 N sulfuric acid was added in each well, and the optical density (OD) was determined by a plate photometer at a wavelength of 450 nm, recorded graphically, and then translated to U/mL.

### 2.3. Leucocytes Immunophenotyping

Blood samples were drawn for evaluation of lymphocyte subsets on day 1 (60 min before the first infusion of RTX), on day 35 (1 week after the last infusion of RTX), and at weeks 24 after the last infusion of RTX. The B-cell surface antigen CD19^+^ was used as a marker for CD20^+^ because the CD20^+^ bound to RTX might interfere with the flow cytometric measurement [15]. The reference range (RR) in percentage for normal values of peripheral blood lymphocyte subsets were: 32–61% for circulating CD4^+^, 14–43% for circulating CD8^+^, and 5–20% for circulating CD19+ B cells [16] Peripheral blood mononuclear cells (PBMCs) were isolated and then stained [17] with monoclonal antibodies anti-CD4 FITC (clone REA623) (Milteny Biotec, Bergisch Gladbach, Germany), anti-CD8 PE (clone REA734), (Milteny Biotec, Bergisch Gladbach, Germany), anti-CD19 APC (clone HIB19) (BD Pharmingen, Milan, Italy), anti-CD20 PC5 (Clone B9E9) (Beckman Coulter, Indianapolis, Indiana USA). Immunophenotyping analysis was performed by multicolor flow cytometry, and cells were collected and analyzed with specific software, as previously reported [18].

### 2.4. Clinical Parameters

The Pemphigus Disease Area Index (PDAI) was developed by the International Pemphigus Committee to separately score disease activity (ulcers/erosions) for skin, scalp, and mucosa membranes, and evidence of damage only for skin and scalp. The total possible maximum score is 250 (120 for skin, 10 for scalp, and 120 for mucosal activity) [19]. Since we only scored oral mucosa, a maximum total score of 90 (excluding the eyes, nose, and anogenital areas) could be obtained.

Complete clinical remission (CCR) “on therapy” and “off therapy”, partial clinical remission (PCR), and “relapse” were defined based on the International Consensus for Pemphigus Vulgaris [20].

### 2.5. Therapeutic Procedure

The OPV patients who were non-responders [14] and/or developed severe side effects related to CIST [20] or relapse during tapering CIST or after CIST were treated with RTX as previously reported [14], after IV premedication with methylprednisolone emisuccinate (20 mg), chlorphenamine maleate (10 mg), and ranitidine hydrochloride (50 mg). Response to treatment was assessed based upon the definitions given in the pemphigus consensus statement [20].

### 2.6. Follow-Up

We performed follow-up visits once a week for the first month, then every other week until CCR was reached, and every 2 months thereafter, recording results from routine blood work with peripheral B-cell count and subsets, and ELISA test for antibodies anti-Dsg 1 and 3, and PDAI score.

### 2.7. Statistical Analysis

Data were analyzed for normality distribution with the Kolmogorov–Smirnov test and were found not normal. The difference in medians of related samples for B and T lymphocytes subpopulations and anti-Dsg-1 and -3 titers at baseline, at the RTX therapy, and 6 months after that were performed by using the Friedman test. If the null hypothesis was rejected, the Wilcoxon matched-pairs signed-rank test with the Bonferroni adjustment was used for multiple comparisons. We also calculated any possible correlation between the clinical and immunological profile at different stages of RTX therapy with Spearman’s correlation coefficient. A heatmap was depicted to highlight the correlations among the analyzed parameters before and after therapy with RTX. A *p*-value of <0.05 was considered significant. IBM-SPSS for Windows software, version 27 (SPSS Inc., Chicago, IL, USA) was used for statistical analyses.

## 3. Results

### 3.1. Clinical Course of Patients

We treated 10 patients with OPV: 3 males and 7 females, with a mean age of 51.7 years (range: 49–57) for males and 42.4 years (range: 30–55) for women. All patients underwent a thorough work-up and, then, initially treated with CIST consisting of high dose of corticosteroid and immunosuppressants (Table 1) [21] for a mean time of 10.5 ± 7.40 months (Figure 1 and Figure 2). Seven patients experienced severe side effects to CIST, of which four also experienced a relapse during tapering or after CIST, and three were also considered non-responder. Of the other three patients without side effects to CIST, two experienced a relapse, and one was considered non-responder.

All OPV patients treated with RTX as adjuvant entered in CCR off therapy and were followed for a mean of 57.2 weeks (range: 23.4 to 157.1 weeks) after the first infusion of RTX. Two patients (20%) experienced mild transient side effects from RTX (tachycardia and headache), and two (patients n° 2 and 4) experienced a relapse [14], treated with an additional 500 mg of rituximab six months after the last infusion [11]. No fatality was observed at the end of the follow-up. 

### 3.2. Serologic Response to RTX Therapy

Serum variations of anti-Dsg-1 and Dsg-3 titer from baseline to 6 months after RTX therapy are depicted in Figure 1A–E and Figure 2A–E for all 10 OPV patients. We observed a significant reduction in anti-Dsg-3 titer during the therapy with RTX, where the median value of serum level of Dsg-3 was 83.50 U/mL at baseline (Q1–Q3: 5.80–184.0), 47.60 U/mL (Q1–Q3: 13.70–153.80) at the end of RTX therapy and 19.35 U/mL (Q1–Q3: 2.00–142.50) 6 months after therapy (*p* < 0.001). However, this decrease in anti-Dsg-3 titer was only significant between anti-Dsg-3 titer at the end of RTX therapy versus baseline (*p* = 0.002) after Bonferroni adjustment (*p* = 0.017).

No substantial variation was noted for anti-Dsg-1 titer, whose median value of its serum level was 5.35 U/mL at baseline (Q1–Q3: 2.00–18.50), 6.60 U/mL (Q1–Q3: 2.00–25.10) at the end of RTX therapy and 2.70 U/mL (Q1–Q3: 2.00–6.40) 6 months after therapy (*p* = 0.085) (Table 2).

### 3.3. Effect of RTX on Lymphocytes Subsets

The percentage of CD4^+^ and CD8^+^ T cells, although varied from baseline to 6 months after RTX therapy, remained within the normal limits after therapy with RTX (Figure 1 and Figure 2): at baseline the median percentage of CD4^+^ was 55.0 (Q1–Q3: 45.0–67.0), which did not vary significantly after therapy with RTX (*p* = 0.081), whereas the percentage of CD8^+^ at baseline (median: 23.0; Q1–Q3: 16.0–25.25) increased significantly at the end of RTX therapy (median: 26.0, Q1–Q3:21.75–28.0) and 6 months after (median: 29.0, Q1–Q3:19.25–34.25) (*p* = 0.009) Similarly, even the CD4^+^/CD8^+^ ratio varied significantly (*p* = 0.011) (Table 2). However, multiple comparisons demonstrated that only the percentage of CD8^+^ 6 months after therapy with RTX versus baseline was significant (*p* = 0.006), as well as the CD4^+^/CD8^+^ ratio 6 months after therapy vs. CD4^+^/CD8^+^ ratio at baseline (*p* = 0.002), after Bonferroni adjustment (*p* = 0.017).

On the other hand, the median of CD20^+^ B-cell percentage was on the lower limit at baseline (median: 5.0; Q1–Q3: 4.0–8.50) and remained significantly reduced for the entire 6 months follow-up (*p* < 0.001) with a very low repopulation in 20% of patients 6 months after the last infusion with RTX (Table 2) (Figure 1A–E and Figure 2A–E). However, this reduction in the percentage of CD20^+^ was only significant between the end of RTX therapy versus baseline (*p* = 0.002) and 6 months after RTX therapy versus baseline (*p* = 0.004), after Bonferroni adjustment (*p* = 0.017).

### 3.4. Correlation between Clinical and Immunological Parameters

The variations of the clinical parameters (PDAI, time to achieve CCR, duration of remission, and time of follow-up) very poorly correlated with immunological ones, except for a slight positive correlation between the anti-Dsg-3 titer at the end of RTX therapy with PDAI score at baseline (*p* = 0.046, *r =* 0.652) and with time to follow-up (*p* = 0.044, *r =* 0.661), and between anti-Dsg-1 titer at CCR with the duration of remission (*p* = 0.042, *r =* 0.659). A stronger correlation was observed between the anti-Dsg-1 titer 6 months after RTX therapy with the duration of remission (*p* = 0.009, *r =* 0.798).

As depicted in Figure 3, the strongest positive correlations were observed within the same cluster of immunological parameters, i.e., anti-Dsg-1 or Dsg-3 titer at different stages of RTX therapy, whereas a strong negative correlation was seen between CD4^+^ and CD8^+^ percentage at different stages of RTX therapy.

## 4. Discussion

The use of RTX in the treatment of PV has been rapidly increasing over the past 15 years, yielding very important results in the attainment of long-lasting remission in the majority of patients that could no longer be treated with CIST [22]. More recently, RTX has also been successfully introduced as either first-line therapy [23] or monotherapy [24].

We treated 10 OPV patients with RTX as an adjuvant following the lymphoma protocol, in combination with a short-term duration of minimal therapy (less than 70 mg of corticosteroids weekly), in order to better control the daily variations in disease activity until RTX begins its therapeutic effects.

We have previously demonstrated that all the OPV patients entered CCR off therapy, experiencing only mild and transitory side effects related to RTX infusions, managed by slowing or briefly interrupting the infusion [14]. Similarly, the rate of relapse of our patients was very low, unlike other studies reporting a relapse rate from 50% [11] to almost 90% [25], probably due to different therapeutic protocols and/or different lengths of follow-up: longer follow-up might result in more relapses.

The immunologic profile of OPV patients showed no correlation between the clinical course of oral disease and anti-Dsg3 titers, as previously confirmed in other reports [11,26]. Indeed, although in all the OPV patients except one, anti-Dsg3 titers gradually and significantly decreased over the observation period, 8 out of 10 patients still presented high anti-Dsg3 Ab levels after RTX therapy, and 6 after 6 months from the end of RTX. Other reports have instead observed a significant positive correlation between anti-Dsg3 titers and severity of oral lesions [27] or between changes in levels of anti-Dsg3 Ab and changes in disease activity for the mucosal component [28]. Interestingly, we also found a weak positive anti-Dsg-1 titer in OPV pts at baseline but not at 6 months follow-up. This positivity seems to be not unusual in clinical practice, as previously reported [29].

However, a more recent study showed that in the long-term period of time, the Dsg3 titer remained at low concentrations in patients treated with rituximab plus short-term prednisone versus patients treated with prednisone alone [23].

On the other hand, it appears that Dsg3 titer strongly predicts relapse by increasing its chance of 28.38 times in patients with mucosal and mucocutaneous involvement [30]; whether this can be true even in patients with exclusive oral involvement is still unknown.

As expected, RTX therapy produced alteration into the lymphocyte subsets. At the baseline, the percentage of CD20^+^ B cells in peripheral blood was close to the lower limit due to the preceding CIST received by all patients before starting RTX. After the first two infusions, the CD20^+^ B cells count was undetectable in all patients and remained as such for the entire follow-up period.

A slow repopulation of CD 20^+^ B cells was seen only in five patients in a median time of 12 months from first RTX infusion (range 7–21 months) (Figure 1 and Figure 2), and two of them experienced a relapse. CD20^+^ B-cell repopulation has been found to be a predictor factor of relapse together with low CD4^+^ T -cell count and positive results of testing for anti-Dsg1 and anti-Dsg3 Ab in a group of 43 mucosal/mucocutaneous PV patients [30]. Whether relapses in OPV patients might be due to the persistence of high titer anti-Dsg 3 antibodies after RTX, where a pathogenic clone could have survived, or simply to new pathogenic clones of B cells secreting pathogenic autoantibodies is yet to be elucidated.

Similarly, patient number 4 experienced a progressive increase in CD20^+^ B-cell count and anti-Dsg3 Ab titers along with a reduction in CD4^+^ T-cell count over the last months of follow-up. Conversely, in patient number two, only a slight repopulation of CD20^+^ B-cell could be detected (Figure 1).

CD4^+^ T regulatory cells have been found to be decreased in PV patients with active disease [31,32]; however, the potential interaction between CD4^+^ T regulatory cells and B cells, and the role of the former in preventing relapses, have not been completely investigated in OPV patients.

Furthermore, we observed a significant increase in CD8^+^ count and CD4^+^/CD8^+^ ratio between baseline and after 6 months from RTX, while no consistent variations were reported in another study on 16 mucocutaneous PV patients [33].

Despite some limitations, as previously reported [14], to the best of our knowledge, this is the first study providing data on the immunological pattern of anti-Dsg3 and anti-Dsg1 Ab, and of leucocyte subsets, in a sample of PV patients with oral involvement, treated with RTX as an adjuvant.

Future research in the OPV patients’ management should be oriented toward the use of RTX as first-line therapy in combination with a low dosage of corticosteroids for a very short-term period, as its early use in PV patients has resulted in a higher response rate [34]. In addition, proper monitoring of OPV patients at regular intervals, including lymphocytes subsets, would be strongly advised in order to identify immunological biomarkers as a potential predictor of clinical relapse also in OPV patients.

## 5. Conclusions

Further prospective studies on larger samples and longer follow-up are needed to confirm our results regarding the role of immunologic parameters in OPV patients treated with RTX. Noteworthy is the current research directed toward the complete elimination of only those autoantibodies produced by autoreactive Dsg 3-specific B cells. The use of targeted therapy with chimeric antigen receptor T cells consisting of Dsg3 fused to CD137-CD3ζ signaling domains (Dsg3 CAAR-T cells) might represent the new frontier in radically treating both mucosal and mucocutaneous PV without the risks of general immunosuppression [35].

## Figures and Tables

**Figure 1 biomolecules-11-01634-f001:**
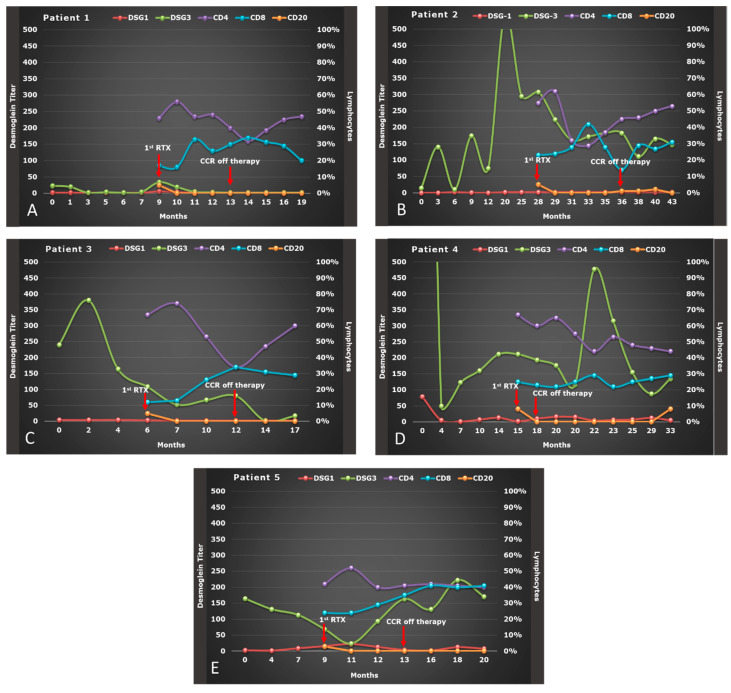
(**A–E**) Evolution of anti-desmoglein (DSG 1 and 3) antibody titer, of CD4, CD8, and CD20, and PDAI-oral mucosa of OPV patients (#1–5) during therapy with RTX.

**Figure 2 biomolecules-11-01634-f002:**
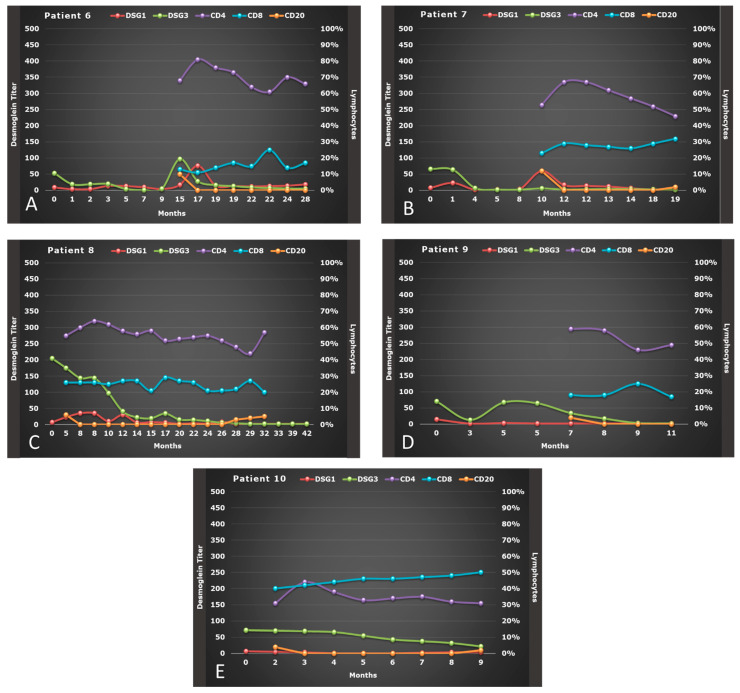
(**A–E**) Evolution of anti-desmoglein (DSG 1 and 3) antibody titer, of CD4, CD8, and CD20, and PDAI-oral mucosa of OPV patients (#6–10) during therapy with RTX.

**Figure 3 biomolecules-11-01634-f003:**
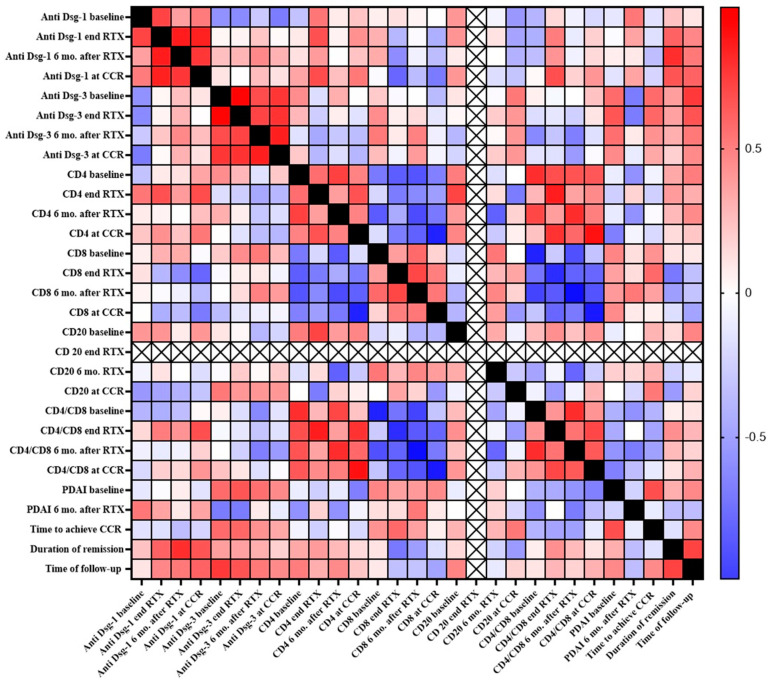
Heatmap of correlations among all the examined clinical and immunological variables. The following correlations were found significant: anti-Dsg-1 baseline vs. anti-Dsg-1 end RTX (p =0.027, r = 0.715); anti-Dsg-1 end RTX vs. anti-Dsg-1 6 mo. after RTX (*p* = 0.005, *r =* 0.860); anti-Dsg-1 end RTX vs. anti-Dsg-1 at CCR (*p* = 0.004, *r =* 0.845); anti-Dsg-1 end RTX vs. CD4 end RTX (*p* = 0.038; r= 0.675); anti-Dsg-1 6 mo. after RTX vs. anti-Dsg-1 at CCR (*p* = 0.015; *r =* 0.767); anti-Dsg-1 6 mo. after RTX vs. duration of remission (*p* = 0.009, *r* = 0.798); anti-Dsg-1 at CCR vs. CD4 end RTX (*p* = 0.032, *r =* 0.688); anti-Dsg-1 at CCR vs. CD4/CD8 end RTX (*p* = 0.035; *r =* 0.682); anti-Dsg-1 at CCR vs. duration of remission (*p* = 0.042, *r =* 0.659); anti-Dsg-3 baseline vs. anti-Dsg-3 end RTX (*p* < 0.001, *r =* 0.952); anti-Dsg-3 baseline vs. anti-Dsg-3 6 mo. after RTX (*p* = 0.033, r = 0.693); anti-Dsg-3 baseline vs. anti-Dsg-3 at CCR (*p* = 0.013, r = 0.770); anti-Dsg-3 end RTX vs. time of follow-up (*p* = 0.015, *r=* 0.758); anti-Dsg-3 end RTX vs. anti-Dsg-3 6 mo. after RTX (*p* = 0.025, r = 0.718); anti-Dsg-3 at CCR vs. anti-Dsg-3 end RTX vs. (*p* = 0.011, r = 0.782); anti-Dsg-3 end RTX vs. PDAI baseline (*p* = 0.046, r = 0.652); anti-Dsg-3 end RTX vs. time of follow-up (*p* = 0.044, r = 0.661); anti-Dsg-3 6 mo. after RTX vs. anti-Dsg-3 at CCR (*p* = 0.003, r = 0.853); CD4 baseline vs. CD4 6 mo. after RTX (*p* = 0.020, r = 0.732); CD4 baseline vs. CD8 end RTX (*p* = 0.048, r = −0.647); CD4 baseline vs. CD8 6 mo. after RTX (*p* = 0.035, r = −0.679); CD4 baseline vs. CD4/CD8 baseline (*p* = 0.008, r = 0.799); CD4 baseline vs. CD4/CD8 end RTX (*p* = 0.035, r = 0.683); CD4 baseline vs. CD4/CD8 6 mo. after RTX (*p* = 0.042, r = 0.661); CD4 baseline vs. CD4/CD8 at CCR (*p* = 0.049, r = 0.646); CD4 end RTX vs. CD4 at CCR (*p* = 0.041, r = 0.665); CD4 end RTX vs. CD20 baseline (*p* = 0.022, *r* = 0.720); CD4 end RTX vs. CD4/CD8 end RTX (*p* = 0.003, r = 0.855); CD4 6 mo. after RTX vs. CD8 baseline (*p* = 0.044, r = −0.657); CD4 6 mo. after RTX vs. CD8 6 mo. after RTX (*p* = 0.025, *r* = −0.711); CD4 6 mo. after RTX vs. CD4/CD8 baseline (*p* = 0.027, r = 0.709); CD4 6 mo. after RTX vs. CD4/CD8 6 mo. after RTX (*p* = 0.007, r = 0.802); CD4 at CCR vs. CD8 at CCR (*p* = 0.001, *r =* −0.889); CD4 at CCR vs. CD4/CD8 end RTX (*p* = 0.010, *r =* 0.781); CD4 at CCR vs. CD4/CD8 at CCR (*p* = 0.001, *r =* 0.896); CD8 baseline vs. CD4/CD8 baseline (*p* = 0.001, *r =* −0.888); CD8 baseline vs. CD4/CD8 6 mo. after RTX (*p* = 0.028, *r =* −0.701); CD8 end RTX vs. CD8 6 mo. after RTX (*p* = 0.025, r= 0.710); CD8 end RTX vs. CD4/CD8 end RTX (*p* = 0.003, *r =* −0.843); CD8 6 mo. after RTX vs. CD4/CD8 baseline (*p* = 0.016, *r =* −0.748); CD8 6 mo. after RTX vs. CD4/CD8 end RTX (*p* = 0.042, *r =* −0.663); CD8 6 mo. after RTX vs. CD4/CD8 6 mo. after RTX (*p* < 0.001, *r =* −0.942); CD8 6 mo. after RTX vs. CD4/CD8 at CCR (*p* = 0.035, *r =* −0.681); CD8 at CCR vs. CD4/CD8 at CCR (*p* < 0.001, *r =* −0.926); CD4/CD8 baseline vs. CD4/CD8 6 mo. after RTX (*p* = 0.007, *r =* 0.802); CD4/CD8 end RTX vs. CD4/CD8 at CCR (*p* = 0.027, *r =* 0.709); PDAI baseline vs. time to achieve CCR (*p* = 0.036, *r =* 0.677); duration of remission vs. time of follow-up (*p* = 0.022, *r =* 0.723). **Dsg**, desmoglein; **RTX**, rituximab; **CCR**, complete clinical remission; **mo.**, months.

**Table 1 biomolecules-11-01634-t001:** Patients’ characteristics.

	No. of Patients (%)
Total	10 (100)
Male	3 (30)
Female	7 (70)
Mean age in years at diagnosis (range)	**M:** 51.7 (49–57) **F:** 42.4 (30–55)
**Drug therapy before therapy with rituximab**	
Corticosteroids and immunosuppressants	8 (80)
Azathioprine	5 (50)
Sulfasalazine	3 (30)
Corticosteroids only	2 (20)
**Clinical remission after therapy with rituximab**	
CCR “off-therapy” after RTX	10 (100)
Transient side effects due to RTX (headache and tachycardia)	2 (20)
Relapse after RTX	2 (20)

**Table 2 biomolecules-11-01634-t002:** Immunological characteristics of oral pemphigus vulgaris patients before and after therapy with rituximab.

**Pts**	**Age**	**Sex**	**Anti-Dsg-1 Baseline ***	**Anti-Dsg-1** **End of RTX**	**Anti-Dsg-1** **6 mo. after RTX**	***p*-Value**	**Anti-Dsg-3 Baseline**	**Anti-Dsg-3** **End of RTX**	**Anti-Dsg-3** **6 mo. after RTX**	***p*-Value**	**CD4/CD8 Baseline**	**CD4/CD8** **End of RTX**	**CD4/CD8** **6 mo. after RTX**	***p*-Value**
1	57	M	6.0	2.0	2.0	0.085	34	3.4	2	<0.001	2.71	1.85	2.35	0.011
2	45	M	2.0	2.0	2.0		308	183	165		2.39	1.32	1.71	
3	55	F	3.0	2.0	2.0		109	67	17.3		5.58	2.04	2.07	
4	44	F	2.0	10.0	5.0		211	193.5	135		2.68	2.61	1.52	
5	28	F	15.4	21.8	7.5		68.3	23.5	170		1.75	2.17	0.98	
6	49	M	17.0	75.8	12.1		97	28.2	8.8		5.23	7.36	4.27	
7	44	F	59.5	13.6	2.0		5.8	3	2		2.30	2.39	1.44	
8	42	F	23.0	35.0	6.0		175	144	22.8		2.12	2.46	2.07	
9	54	F	2.0	2.0	2.0		34.2	17.1	2		3.28	3.22	2.88	
10	30	F	4.7	3.2	3.4		70	68.5	21.4		0.78	1.05	0.62	
**Pts**			**Cd4 (%)** **Baseline**	**CD4 (%)** **End of RTX**	**CD4 (%)** **6 mo. after RTX**	***p*-Value**	**CD8 (%)** **Baseline**	**CD8 (%)** **End of RTX**	**CD8 (%)** **6 mo. after RTX**	***p*-Value**	**CD20 (%)** **Baseline**	**CD20 (%)** **End of RTX**	**CD20 (%)** **6 mo. after RTX**	***p*-Value**
1			46	48	47	0.081	17	26	20	0.009	5	0	0	<0.001
2			55	37	53		23	28	31		5	0	0	
3			67	53	60		12	26	29		5	0	0	
4			67	60	44		25	23	29		8	0	8	
5			42	52	40		24	24	41		3	0	0	
6			68	81	64		13	11	15		10	0	0	
7			53	67	46		23	28	32		12	0	2	
8			55	64	56		26	26	27		6	0	0	
9			59	58	49		18	18	17		4	0	0	
10			31	44	31		40	42	50		4	0	2	

**M**, male; **F**, female; **RTX**, rituximab; **Dsg**, desmoglein; ***** desmoglein titer is measured in U/mL.

## Data Availability

Data available on request due to privacy restrictions. The data presented in this study are available on request from the corresponding author.

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
