# Peer review of "Immunocytometric Analysis of Oral Pemphigus vulgaris Patients after Treatment with Rituximab as Adjuvant"

_biomolecules, 2021, doi:10.3390/biom11111634_

Round 1
Reviewer 1 Report
Authors report the study on the immunological pattern of anti-Dsg3 and anti- Dsg1 Ab, and of leucocyte subsets, in a sample of pemphigus vulgaris patients with exlusively oral involvement, treated with Rytuximab as adjuvant. They demonstrated a significant impact on anti Dsg-3 titer and B and T cell lymphocyte subpopulations profile.
It has been shown by Amagai et.at (Am Dermatol 1999) that the clinical phenotype of pemphigus is defined by the anti-desmoglein autoantibody profile and oral pemphigus vulgaris is associated with anti-desmoglein-3 but not with anti-desmoglein-1 antibody.
Author should clarify why some of the patients with oral pemphigus vulgaris included to this study presented significant titer of anti-desmoglein-1 antibodies.
Author Response
Reviewer 1
Authors report the study on the immunological pattern of anti-Dsg3 and anti- Dsg1 Ab, and of leucocyte subsets, in a sample of pemphigus vulgaris patients with exclusively oral involvement, treated with Rituximab as adjuvant. They demonstrated a significant impact on anti Dsg-3 titer and B and T cell lymphocyte subpopulations profile.
It has been shown by Amagai et.at (Am Dermatol 1999) that the clinical phenotype of pemphigus is defined by the anti-desmoglein autoantibody profile and oral pemphigus vulgaris is associated with anti-desmoglein-3 but not with anti-desmoglein-1 antibody.
Author should clarify why some of the patients with oral pemphigus vulgaris included to this study presented significant titer of anti-desmoglein-1 antibodies.
RESPONSE: thank you for this comment. It is not unusual in clinical practice to see oral PV pts with positive anti-DSG-1 titer, as previously reported (Sardana K, et al Br J Dermatol. 2013 Mar;168(3):669-74) We added this into discussion section.
Reviewer 2 Report
Fortuna et al. performed an immuoncytometric analysis of oral pemphigus vulgaris patients who had been treated with Rituximab. If I understood correctly, the paper is an analytic continuation of their work from Fortuna et al. 2020 (reference 14) involving the same patient cohort.
The authors measured Dsg-1 and -3 via ELISA and quantified CD4+, CD8+ and CD19+ cells via FACS analysis. Overall, the manuscript does not contain many data and the Figures are difficult to look at.
Major comment:
Recently, immunphenotyping of PV samples was published by a different group (Holstein et al.) where the importance of TH17 cells in PV and IL-17 was described. The authors should perform some more immunophenotyping analyses on T and B cell subtypes to get a more complete picture (e.g. Th17 cells, TFH17 cells, IL17 and other cytokines, memory B cells)
Minor comments:
- In the materials and methods section the “Leucocyte immunophenotyping” should be explained in more detail: how were PBMCs isolated and how was the staining performed?
- Line 104: was the dilution really 1:101?
- Figures 1 and 2: the Figures are difficult to look at. The X-axis is different in most graphs. Instead of only showing the lines, individual data points should be shown
- in addition to the 10 separate graphs, a summary graph (e.g. boxplot) should be prepared
- Table 2 should be reformatted to fit on one page
- Reference #14 and 20 seem to be identical?
Author Response
Reviewer 2
Fortuna et al. performed an immunocytometric analysis of oral pemphigus vulgaris patients who had been treated with Rituximab. If I understood correctly, the paper is an analytic continuation of their work from Fortuna et al. 2020 (reference 14) involving the same patient cohort.
The authors measured Dsg-1 and -3 via ELISA and quantified CD4+, CD8+ and CD19+ cells via FACS analysis. Overall, the manuscript does not contain many data and the Figures are difficult to look at.
Major comment:
Recently, immunphenotyping of PV samples was published by a different group (Holstein et al.) where the importance of TH17 cells in PV and IL-17 was described. The authors should perform some more immunophenotyping analyses on T and B cell subtypes to get a more complete picture (e.g. Th17 cells, TFH17 cells, IL17 and other cytokines, memory B cells)
RESPONSE: thank you for your comment. However, we have underlined in our materials and methods that this is a retrospective study: it cannot be possible to perform further analyses, as samples are no longer available.
Minor comments:
In the materials and methods section the “Leucocyte immunophenotyping” should be explained in more detail: how were PBMCs isolated and how was the staining performed?
RESPONSE: we added the appropriate reference regarding the methodology (Romano C, et al. Leuk Lymphoma. 2003 Nov;44(11):1963-71.)
Line 104: was the dilution really 1:101?
RESPONSE: Yes, it was
Figures 1 and 2: the Figures are difficult to look at. The X-axis is different in most graphs. Instead of only showing the lines, individual data points should be shown
RESPONSE: Figures are difficult to look at because the system did not allow us to upload them as a separate bmp file in high resolution but just as merged within the word/pdf file. The X-axis cannot be identical in all pts as they entered in remission at different time points. To make the graphs more readable we had to adapt the X-axis to the individual pt’s remission. We added data point on the lines and changed the colors in an attempt to make them clearer.
In addition to the 10 separate graphs, a summary graph (e.g. boxplot) should be prepared
RESPONSE: An additional boxplot containing data from all pts would be redundant and confusing since already present in Table 2. Diagrams are detailed and self-explanatory.
Table 2 should be reformatted to fit on one page
RESPONSE: we apologize for this inconvenience, but this was due to the conversion of the word file into pdf. In our word file, Table 2 fits in 1 page. We hope that the editorial office solved this issue.
Reference #14 and 20 seem to be identical?
RESPONSE: We deleted one of the 2 references and reformatted accordingly
Round 2
Reviewer 1 Report
The article was improoved
